# Effect of Bacterial Extracellular Polymeric Substances from *Enterobacter* spp. on Rice Growth under Abiotic Stress and Transcriptomic Analysis

**DOI:** 10.3390/microorganisms12061212

**Published:** 2024-06-16

**Authors:** Yosra Aoudi, Shin-ichiro Agake, Safiullah Habibi, Gary Stacey, Michiko Yasuda, Naoko Ohkama-Ohtsu

**Affiliations:** 1United Graduate School of Agriculture, Tokyo University of Agriculture and Technology, 3-5-8 Saiwaicho, Fuchu-shi 183-8509, Tokyo, Japan; 2Institute of Global Innovation Research, Tokyo University of Agriculture and Technology, 3-8-1 Harumicho, Fuchu-shi 183-8538, Tokyo, Japan; nohtsu@cc.tuat.ac.jp; 3Division of Plant Science and Technology, University of Missouri—Bond Life Sciences Center, 1201 Rollins St., Columbia, MO 65201-4231, USA; 4Institute of Agriculture, Tokyo University of Agriculture and Technology, 3-5-8 Saiwaicho, Fuchu-shi 183-8509, Tokyo, Japan

**Keywords:** extracellular polymeric substances, abiotic stress, RNA-seq analysis, gene ontology analysis, *Enterobacter* spp., plant biostimulants

## Abstract

Plant biostimulants have received attention as sustainable alternatives to chemical fertilizers. Extracellular polymeric substances (EPSs), among the compounds secreted by plant growth-promoting rhizobacteria (PGPRs), are assumed to alleviate abiotic stress. This study aims to investigate the effect of purified EPSs on rice under abiotic stress and analyze their mechanisms. A pot experiment was conducted to elucidate the effects of inoculating EPSs purified from PGPRs that increase biofilm production in the presence of sugar on rice growth in heat-stress conditions. Since all EPSs showed improvement in SPAD after the stress, *Enterobacter ludwigii*, which was not characterized as showing higher PGP bioactivities such as phytohormone production, nitrogen fixation, and phosphorus solubilization, was selected for further analysis. RNA extracted from the embryos of germinating seeds at 24 h post-treatment with EPSs or water was used for transcriptome analysis. The RNA-seq analysis revealed 215 differentially expressed genes (DEGs) identified in rice seeds, including 139 up-regulated and 76 down-regulated genes. A gene ontology (GO) enrichment analysis showed that the enriched GO terms are mainly associated with the ROS scavenging processes, detoxification pathways, and response to oxidative stress. For example, the expression of the gene encoding *OsAAO5*, which is known to function in detoxifying oxidative stress, was two times increased by EPS treatment. Moreover, EPS application improved SPAD and dry weights of shoot and root by 90%, 14%, and 27%, respectively, under drought stress and increased SPAD by 59% under salt stress. It indicates that bacterial EPSs improved plant growth under abiotic stresses. Based on our results, we consider that EPSs purified from *Enterobacter ludwigii* can be used to develop biostimulants for rice.

## 1. Introduction

Global climate change and environmental stresses are major challenges that limit crop productivity and threaten worldwide food security [1]. With the continuous increase in world population, which is expected to reach nearly 10 billion people by 2050 [2], farmers still rely on chemical fertilizers to increase yields and fulfill the world food demand. However, the long-term use of these chemicals will disturb the plant–soil–microbe system. It will negatively change the biomass and diversity of soil microbial communities [3]. Thus, empowering sustainable solutions in agriculture is becoming a necessity. Plant biostimulants (PBs) have received much attention in modern agriculture as an alternative to synthetic chemicals [4]. Europe represented the largest market for biostimulants in 2022 [5]. By 2030, the global biostimulants market is expected to reach USD 7.97 billion from USD 3.69 billion in 2023 [6]. We can distinguish two major categories of plant biostimulants: substances and microorganisms. The first category covers humic substances, including humic and fulvic acids, protein hydrolysates and amino acids, biopolymers, inorganic compounds, and seaweed extracts. Beneficial bacteria, principally plant growth-promoting rhizobacteria (PGPRs), and beneficial fungi form the category of microorganisms [7,8].

PGPRs are a group of plant-associated microbes that colonize the plant rhizosphere. Many studies have shown their major role in stimulating plant growth and mitigating abiotic stresses in crops. Hence, PGPRs represent the best eco-friendly approach to be adopted, especially since abiotic stresses are expected to increase due to climate change [8]. Growth promotion by microbial plant stimulants involves direct and indirect modes of action. Direct mechanisms include nitrogen fixation, phosphate solubilization, potassium solubilization, iron sequestration, and phytohormone production. In contrast, indirect mechanisms include antibiosis, lytic enzyme production, competition, induced systemic resistance (ISR), and exopolysaccharide production [9].

Extracellular polymeric substances (EPSs) are hydrated biopolymers excreted by microorganisms and contribute to biofilm formation. EPSs comprise mainly polysaccharides, proteins, nucleic acids, and lipids; they create an optimal framework to bolster the structural integrity and stability of biofilms by, e.g., facilitating their attachment to surfaces, protecting barriers, aggregating their cells, retaining moisture, and serving as nutrient sources [10]. A main component of EPSs are polysaccharides, which can chelate harmful ions (e.g., Na+), regulate water availability, and improve soil structure, contributing to enhanced stress tolerance. They facilitate the formation of biofilms and the attachment of beneficial bacteria to plant roots, protecting them against biotic and abiotic stresses [11,12].

The available literature focuses more on the effects of EPS-producing bacteria rather than purified EPSs on plant growth and stress tolerance without delving into the specific molecular mechanisms and gene expression patterns involved in those responses. Thus, the current study attempts to explore the effects of purified bacterial EPSs on plant growth and to identify the responsive genes implicated in plant growth promotion and stress tolerance by exogenous application of EPSs on rice seeds through RNA sequencing and transcriptome analysis.

## 2. Materials and Methods

### 2.1. Screening for EPS-Producing PGPR

Previously, we isolated PGPRs from rice plants and investigated their PGR activity, such as indole-3-acetic acid (IAA) production, acetylene reduction activity (ARA), and phosphate solubilization activity [13]. In the current study, 17 bacterial strains among 30 isolates were selected based on differences in the genera (*Pseudomonas*, *Rhizobium*, *Stenotrophomonas*, *Agrobacterium*, *Enterobacter*, *Bacillus*, and *Xanthomonas*), the type of rhizosphere soil they were derived from, their ability to produce IAA, and their ARA activity, in addition to their positive effects on rice biomass [14]. To investigate the proper medium for EPS production, isolates were grown in 2 different media: tryptic soy broth (TSB) and yeast malt extract broth (YM Broth). 

The growth of the isolates was determined by measuring optical density (OD) at 600 nm. The screening for EPS production activity in vitro consisted of placing a colony on Petri plates containing the previously quoted medium without or with 3% (*w*/*v*) glucose. After incubation at 28 °C for 48 h, the production of EPSs was apparent by the formation of a white mucoid appearance around the colonies. In addition, EPS production was detected in the presence of 0.1% Fluorescent Brightener 28 in the YM agar medium. The EPS matrix was then visualized by fluorescent microscopy (Olympus BX50, Tokyo, Japan) using an omega optical filter set (XF88-2).

### 2.2. EPS Purification and Measurement of Yields

EPSs were extracted and purified from 6 strains following the protocol of Ortega et al. [14] with some modifications. A pure colony of each selected bacteria was cultured in 100 mL of YM broth with 3% (*w*/*v*) glucose and incubated at 30 °C in a rotary shaker at 152 rpm for 32 h. After heating at 100 °C for 15 min, the supernatant was obtained through centrifugation at 25 °C and 10,000 rpm for 10 min. The samples were filtered through a 0.20 µm filter. Two volumes of 95% cold ethanol were added to the supernatant and then incubated overnight at 4 °C. The precipitated EPS was obtained through centrifugation at 25 °C and 10,000 rpm for 10 min and then redissolved in 10 mL of deionized water. The samples were dialyzed (Mw cut-off 3500 Da) using a Spectra/Por^®^ dialysis molecular porous membrane (Spectrum Laboratories, Inc., Rancho Dominguez, CA, USA) to remove impurities for 48 h by changing the water once. The crude EPS was concentrated by freeze-drying for 72 h. After measuring the final yield (mg/L), the purified EPS of each strain was dissolved in deionized water. We quantified some specific fractions of interest in the crude EPS by different colorimetric methods. The anthrone–sulfuric acid method estimated the total carbohydrate content with glucose as standard [15]. The dinitrosalicylic acid (DNS) reagent was adopted to determine reducing sugars [16]. Protein estimation was conducted following the instructions of the TAKARA BCA Protein Assay Kit (TaKaRa Bio, Inc., Shiga, Japan) with bovine serum albumin (BSA) as standard. EPS solutions were stored at 4 °C until further use.

### 2.3. Plant Assay under Heat Stress and Optimization of EPS Concentration for Inoculation

To eliminate possible contamination, rice seeds were surface sterilized with 70% ethanol for 1 min, followed by immersion in 1% (*v*/*v*) sodium hypochlorite (NaClO) for 15 min and washed five times with sterile distilled water. The sterilized seeds were soaked in sterile distilled water and incubated for pre-germination at 28 °C for 48 h. The pre-germinated rice seeds were sown within 1 cm of the surface in plastic plant pots filled with 300 g of sterilized low-nutrient Inaho-baido soil (INAHO-KAKO CORPORATION, Toyama, Japan), autoclaved beforehand at 121 °C for 30 min. Inaho-baido soil is a semi-granular, sandy clay with a pH range of 4.5 to 5.5. Additionally, it is deficient in nitrogen (N), phosphorus (P), and potassium (K) nutrients. Ten seeds in each cultivation pot were inoculated with three different concentrations of EPS solution (1 µg, 10 µg, and 100 µg/mL) extracted previously from 6 selected bacterial strains. For the control, the seeds were inoculated with sterile distilled water of the same volume for the same period. All treatments were conducted in triplicate. For heat stress, the seedlings were kept in a growth chamber with constant illumination at 25 °C with a photoperiod of 16/8 h light/dark cycle for ten days. Heat stress was then applied at 45 °C for 6 h during the light cycle for 4 days. Seedlings were subsequently incubated with the previous conditions at 25 °C for two days before harvesting. Different growth parameters were measured: plant establishment rate, fresh and dry weight of shoots, and fresh and dry weight of roots. The plant establishment corresponds to the number of seedlings with shoot lengths over 7 cm [17]. Shoot lengths were measured before and after heat stress. Leaf chlorophyll content was estimated by checking SPAD values before and after heat stress using SPAD-502Plus (Konica, Minolta, Tokyo, Japan). Shoots and roots dry weights were checked after drying plant materials at 80 °C for 48 h.

### 2.4. RNA Extraction from Rice Embryos

Total RNA was extracted from rice embryos following the instructions of the RNeasy Plant Mini Kit (Qiagen, Hilden, Germany) with some modifications. The rice seeds were first surface sterilized with the same procedure above. The surface sterilized seeds were soaked in a beaker filled with sterile distilled water and incubated at 25 °C for 24 h. Around 30 seeds were used in each replication for both groups: the mock treatment, where the seeds were imbibed with sterile water, and the EPS treatment, where the seeds were imbibed with 10 µg/mL of EPS solution in Petri dishes between 2 filter papers and incubated at 25 °C for 24 h. Three replicates were used for mock and EPS treatment each. Around 100 mg of embryos of the rice seeds from each replicate were transferred to a mortar and homogenized in liquid nitrogen. As the rice seeds contain large quantities of polysaccharides, 1 mL of Fruit-mate^TM^ (TaKaRa Bio, Inc., Shiga, Japan) was added to the homogenized plant tissue in a 1.5 mL RNase-free tube, cooled in advance in liquid nitrogen. The homogenized tissue and Fruit-mate^TM^ were mixed and centrifuged immediately at 12,000× *g* at 4 °C for 5 min. The supernatant was transferred and divided equally into two new 1.5 mL RNase-free tubes, 0.5 mL of RNiso Plus was added to each tube, and the homogenate was mixed and stood at room temperature for 5 min. Subsequent steps were according to the RNeasy Mini Kit (Qiagen). At the end, the RNA was eluted in 50 µL of RNase-free water.

### 2.5. RNA Sequencing (RNA-Seq) and Data Analysis

RNA samples from three biological replicates in both groups (control and treatment) were then sent to the Beijing Genomics Institute (BGI, Shenzhen, China) for RNA-seq analysis using the DNBSEQ platform. To meet the requirements of library construction and sequencing, the quality of the samples was first assessed using the Agilent Bioanalyzer 2100 system (Agilent Technologies, Santa Clara, CA, USA). To quantify gene expression levels, the gene amount was later calculated under three different Fragments Per Kilobase of transcript per Million mapped reads ranges FPKM (FPKM ≤ 1, FPKM 1~10, and FPKM ≥ 10). The online-based software iDEP2.0 [18] was used to identify the differentially expressed genes (DEGs) following the DESeq2 method by adjusting the threshold of false discovery rate (FDR cut-off) to less than 0.05 and min fold-change equal to 1. The volcano plot, the principal component analysis PCA, and the K-means clustering analysis were performed by the same online software. The identified DEGs were uploaded to the online web-based software ShinyGO v0.741 and DAVID for Gene Ontology (GO) and Kyoto Encyclopedia of Genes and Genomes (KEGG) pathway analyses.

### 2.6. Estimation of Gene Expression through Quantitative Real-Time PCR (qPCR)

Total RNA extracted from rice embryos with the RNAiso Plus Kit (TaKaRa Bio, Inc.) was reverse transcribed using the PrimeScriptTM RT Reagent Kit with gDNA Eraser (Perfect Real Time) (TaKaRa Bio, Inc.) following the manufacturer’s instructions. Quantitative RT-PCR was performed on LightCycler^®^ 96 System Version 2.0 (Roche Diagnostics GmbH, Mannheim City, Germany) using KAPA SYBR^®^ FAST qPCR Master Mix (2X) Kit (Kapa Biosystems Inc., Wilmington, MA, USA). Each reaction contained 5 µL of 2× KAPA SYBR FAST qPCR Master Mix, 0.2 µL of 10 µM forward primer, 0.2 µL of 10 µM reverse primer, 1 µL of complementary DNA (cDNA), and 3.6 µL of PCR-grade water. The following reaction procedure was undertaken: stage 1—pre-incubation, 95 °C for 180 s; stage 2—3-step amplification, 95 °C for 10 s, 57 °C for 20 s, and 72 °C for 1 s for 45 cycles; stage 3—melting, 95 °C for 5 s, 65 °C for 60 s, and 97 °C for 1 s; and stage 4—cooling, 40 °C for 10 s. All primer sequences for standard and target genes are shown in Appendix A. Relative gene expression levels were evaluated using the ΔΔCT method [19]. The rice *ubiquitin* gene was used as an internal control to normalize the expression levels of the target gene *OsAAO5* [20].

### 2.7. Effect of EPSs from Enterobacter ludwigii on Rice under Abiotic Stresses

The abiotic stress assay was conducted with 3 different conditions other than normal conditions: salinity, drought, and heat stress. For EPS treatment in all conditions, the pre-germinated rice seeds were inoculated with 10 µg of the EPS produced by the strain JW191 and with sterile water for control plants. Regarding salt stress, the pre-germinated rice seeds were transplanted in soil (sterilized low-nutrient soil Inaho-baido (INAHO-KAKO CORPORATION, Toyama, Japan) containing 150 mM NaCl and incubated at 25 °C with a photoperiod of 16 h light/8 h dark cycle for 16 days. Irrigation with sterile water purified with a reverse-osmosis membrane (RO water) was maintained regularly every 48 h. In addition, salt solutions similar to initial concentrations were used for irrigation 3 times on day 0, day 7, and day 14 after sowing. For drought stress, seedlings were initially grown under a normal watering regime every 48 h for 9 days. After that, watering was halted for 7 days until harvesting. For heat stress, the seedlings were kept in a growth chamber with constant illumination at 25 °C with a photoperiod of 16 h light/8 h dark cycle for 10 days. After that, heat stress was applied at 45 °C from 10 a.m. to 4 p.m. for 4 days. After heat stress treatment, seedlings were incubated under normal conditions at 25 °C for 2 days before harvesting. Seedlings were harvested for all treatments, and different growth parameters were measured, i.e., plant establishment rate, shoot length, fresh and dry weight of shoots, fresh and dry weight of roots, and chlorophyll content (SPAD).

### 2.8. Statistical Analysis

To investigate the differences observed among the strains and EPS concentrations, analysis of variance (ANOVA) and Tukey post hoc tests for multiple comparisons were performed using SPSS Statistics v. 23 (IBM SPSS Statistics, Armonk, NY, USA). A *t*-test was also performed using Microsoft Excel Office 365.

## 3. Results

### 3.1. Screening of EPS-Producing PGPRs

Initially, the 17 candidate strains were grown in 2 media: tryptic soy broth (TSB) and yeast malt extract (YM) broth at 28 °C for overnight incubations. According to the optical density (OD) at 600 nm, TSB supported bacterial growth better than the YM medium (Table 1). A total of 0.1% of Fluorescent Brightener 28 was added to both agar mediums without or with a 3% (*w*/*v*) glucose for visualization of the EPS matrix under fluorescence microscopy. Fluorescent Brightener 28 is a fluorochrome that binds specifically to cellulose and chitin, which are common components of EPSs [21]. Under the fluorescence microscope, EPSs produced by the strains JM160, JW191, JM187, JC20, JO32, and JM63 appeared brightly fluorescent when glucose was added to the YM agar medium. The other strains did not show the ability to produce EPSs in vitro, either without or with 3% (*w*/*v*) glucose in either medium (Appendix A). Thus, YM agar was favored for the production of EPSs by the isolates in the presence of sugar. Based on bacterial growth and the ability to produce EPSs in vitro, the selected isolates (JM160, JW191, JM187, JC20, JO32, and JM63), which belong to *Enterobacter* spp., were used for further experiments (Table 1).

In addition, the results of EPS production showed that the produced EPS varied significantly among the same PGPR genera, with JC20 the highest EPS producer and JW191 the lowest (Figure 1). The EPS was quantified for its composition in total carbohydrates, total protein content, and reducing sugar content. The results showed that the average composition was 28% carbohydrates, 10% proteins, and 3% reducing sugars.

### 3.2. Effect of EPS Treatment on Rice Growth under Heat Stress

A range of EPS concentrations purified from the selected isolates were used to evaluate the comparative effect on plant growth under heat stress. Plant growth parameters were measured, such as the shoot length, the fresh and dry weight of shoots and roots, and the SPAD values (Table 2).

Overall, EPS treatment caused a general enhancement of plant biomass under heat stress. As compared to non-inoculated seedlings, 100 µg of EPS from strains JM160 and JM63 significantly improved shoot length by 29% after heat stress. In addition, 10 µg of EPS from the strain JW191 was sufficient to increase the shoot fresh weights and lengths of seedlings by 14%. The same percentage increase was obtained for root dry weight in seedlings inoculated with 10 µg from strain JM187. Notably, a significant increase in SPAD values was more obvious in seedlings inoculated with 10 µg of EPS regardless of the strain of origin. The results showed that the chlorophyll content increased by 40% and 50% in heat-stressed plants inoculated with 10 µg of EPS from the strains JW191 and JC20, respectively, compared to control plants. Based on these results, 10 µg of EPS appears sufficient to enhance plant growth under heat stress.

### 3.3. Transcriptome Analysis of EPS Treatment of Rice Seeds

In order to gain some insight into the effects of EPS treatment on rice, we performed a comparative transcriptome analysis using rice seedlings 24 h after germination. This time point represents a key developmental stage in rice seed germination. It is characterized by physiological changes such as starch degradation, sucrose utilization, and increased amylase activity, as well as molecular changes involving the up-regulation of genes related to alpha-amylase synthesis, gibberellin metabolism, and hormonal regulation, which collectively initiate and promote the germination process [22,23]. Total RNA was extracted from rice seed embryos inoculated previously with 10 µg of EPS obtained from the strain JW191 or water for 24 h with 3 replicates in each group. We sequenced 6 samples using the DNBSEQ platform, generating ~6.81 G Gb bases per sample. The average mapping ratio with the reference genome was 98.03% and 83.65% gene coverage; 33,147 genes were identified (Appendix A). To reflect the correlation of gene expression between samples, the Pearson correlation coefficients of all gene expressions between each two samples were calculated, and these coefficients were reflected in the form of a heatmap. The correlation coefficients can reflect the similar situation of the overall gene expression between each sample. The higher the correlation coefficient, the more similar the gene expression level (Appendix A). A bar plot of total raw read counts per library was generated (Appendix A). The *x*-axis displays the six samples for both groups (control and EPS treatment), while the *y*-axis represents the total raw read counts measured in millions of reads. Each bar corresponds to one replicate, with heights indicating the total number of raw reads obtained. The read counts of each sample were obtained in similar numbers. Furthermore, the distribution of the transformed data showed a small variation among replicates (Appendix A). The scatter plot in Appendix A illustrates the correlation between the two conditions (control and EPS treatment) after applying log transformation to the expression values. The *x*-axis displays the log-transformed expression values for control, while the *y*-axis represents the log-transformed expression values for EPS treatment. Each point on the scatter plot corresponds to a specific gene, with its position determined by its expression levels in the two conditions. The scatter plot shows a clear positive correlation between the expression levels in both conditions. Most genes clustered around the diagonal line, suggesting similar expression patterns in both conditions, while a subset of genes is differentially regulated. Principal component analysis PCA plot using the most variation PC1 and the second most variation PC2 revealed a clear difference between control and treatment. RNA seq data showed that 71.3% of the variation in the transcriptome data was due to EPS treatment, while 28.7% of the variation was not due to treatment (Appendix A). With the DESeq2 method, we identified 139 up-regulated and 76 down-regulated differentially expressed genes (DEGs), respectively, using a threshold of false discovery rate (FDR cut-off) less than 0.05 and min fold-change equal to 1.0. The volcano plot indicates that EPS treatment led to a massive transcriptional response (Figure 2a). Ascorbate oxidase gene *OsAAO5* (Os09g0507300), which acts as an antioxidant under stress conditions, was the most highly enriched by EPS treatment with a positive fold change (log2FC) of 5.45 (Figure 2a). K-means clustering was used to divide the top 250 genes of all available gene sets into 6 clusters based on the similarity of their gene expression patterns (Appendix A). Cluster 4, which showed substantial differences between treated and untreated samples, included *OsAAO5* (Figure 2b). A heatmap was then created to show the expression of genes in significant gene sets (Appendix A). The expression of *OsAAO5* was validated by qRT-PCR analysis. Figure 2c indicates that the relative gene expression of the *OsAAO5* gene was 2 times higher in the treated seeds compared to non-treated seeds.

The up- and down-regulated DEGs were subjected to gene ontology enrichment analysis (GO). GO was used to evaluate the potential functions of significantly enriched DEGs by EPS treatment, where they were categorized into biological processes (GO_BP), cellular components (GO_CC), and molecular functions (GO_MF). GO analysis indicates the enrichment of important biological processes associated with up-regulated DEGs, such as reactive oxygen species (ROS) metabolic process (GO:0072593), cellular response to toxic substances (GO: 0097237), and detoxification process (GO: 0098754). The molecular functions associated with the up-regulated DEGs mainly focused on oxidoreductase activity (GO: 0016491) and heavy metal binding (GO: 0046872). In terms of cellular components, the up-regulated DEGs were enriched in the extracellular region (GO: 0005576) and cell periphery (GO: 0071944) (Figure 3a, Appendix A). In addition, the results of GO analysis revealed that down-regulated DESs were significantly enriched in biological processes, including the regulation of several transport processes such as amino acids and organic acids. The response to heat (GO:0009408) and the response to temperature stimulus (GO:0009266) pathways were also significantly enriched by down-regulated DEGs (Figure 3b, Appendix A). No significant enrichments were found in the molecular functions and cellular components associated with down-regulated DEGs.

Kyoto Encyclopedia of Genes and Genomes (KEGG) pathway enrichment analysis was also conducted. The KEGG results revealed that the up-regulated DEGs were highly associated with several metabolism pathways, including linoleic acid and tyrosine metabolisms, ubiquinone and other terpenoid–quinone biosynthesis, beta-alanine and pyruvate metabolisms (Figure 4a), whereas the down-regulated DEGs were enriched in galactose metabolism, citrate cycle, ubiquitin-mediated proteolysis, and glycolysis (Figure 4b). GO analysis revealed that *OsAAO5* was enriched in several pathways related to molecular functions such as oxidoreductase activity (GO:0016491), cation binding (GO:0043169), metal ion binding (GO:0046872), transition metal ion binding (GO:0046914), and copper ion binding (GO:0005507). This gene was also enriched in the extracellular region (GO:0005576) (Appendix A).

### 3.4. Effect of EPS Inoculation on Abiotic Stresses

We used the EPS produced by JW191 as a bioinoculant to confirm its effect on the growth parameters of rice plants under normal and stress conditions (drought, salinity, and heat). As compared with normal conditions without EPS treatment, these stresses severely affected plant biomass. For example, shoot and root fresh weights decreased by 67% and 50%, respectively, under drought, by 43% and 35% under heat stress, and by 44% and 47% under 150 mM salinity (Appendix A). The application of EPS as a bioinoculant has been shown to contribute to an overall biomass improvement in stressed plants compared to control plants (Table 3). For instance, EPS inoculation increased shoot fresh weight by 20% under heat. Root fresh weight increased by 47% and 32% under heat and drought, respectively, with EPS treatment. The statistical analysis revealed that EPS treatment was more effective under heat stress as it showed significant increases in all measured growth parameters for heat-stressed plants compared to control plants. Under drought stress, EPS treatment positively affected plant establishment, shoot and root dry weights, and SPAD values. Moreover, with a higher concentration of NaCl (150 mM), EPS inoculation was effective on plant establishment and SPAD value but not for other growth parameters (Table 3).

## 4. Discussion

In the present study, we compare the effectiveness of bacterial extracellular polymeric substances (EPSs) purified from different strains on plant growth and development of japonica rice (*Oryza sativa* L.) under abiotic stresses. The first screening of EPS-producing PGPRs showed that 6 isolates appeared to produce EPS in vitro in the presence of sugar compared to other bacteria. These strains were used for further experiments. Surprisingly, the selected isolates all belong to the genus *Enterobacter*. Many researchers have noted the ability of *Enterobacter* to produce EPSs relative to other bacteria. *Enterobacter* spp. likely possess unique genetic and biochemical features to allocate more resources toward polysaccharide synthesis than other bacteria [24,25]. Our results showed that EPS production can vary across bacterial strains, which may lead to diverse structural and functional properties that impact plant growth and stress tolerance. We observed that EPS inoculation led to higher shoot length and weight and, ultimately, improved plant biomass in seedlings subjected to abiotic stresses like drought, salinity, and heat. Previously, it was demonstrated that the applications of extracellular polymeric substances (EPSs), originating from *Pseudomonas chlororaphis* O6, to epidermal peels of leaves of *Arabidopsis thaliana* led to the induction of systemic drought tolerance [26]. Within the same framework, Sun et al. clarified in two successive studies the function of polysaccharides by constructing a deficient strain obtained from *Pantoea alhagi* NX-11, and comparing the effect on rice salt resistance. They found that rice seedlings treated with polysaccharides and those treated with fermentation broth of wild strain grew better than those treated with the fermentation broth of EPS deficient strains or in the control group. Their findings suggested that the wild strain could boost the salt stress tolerance of rice with polysaccharides playing a pivotal role in the quoted mechanism [27,28]. Another study found that applying EPSs extracted from *Bacillus* sp. S3 during cadmium stress resulted in increased biomass in rice (*Oryza sativa* L.) [29]. Likewise, it was reported that *Citrobacter* sp. contributed to salinity stress mitigation in *Vingna radiata* (L.) by producing EPSs and forming biofilms [30]. In line with this, we found that the ethanol-precipitated EPS from *Enterobacter ludwigii* JW191 could directly enhance plant growth and stress tolerance, even without the presence of the bacterial cells themselves. This demonstrates the direct EPS-mediated effects on plant growth promotion and mitigation of stress.

RNA-seq analysis was used to study the transcriptomic changes in rice embryos in response to inoculation with 10 µg of EPS from the strain *Enterobacter ludwigii* (JW191). According to the previous study conducted by Habibi et al. [13], the isolate JW191 increased the shoot length, root length, shoot dry weight, and root dry weight of rice plants by 19%, 68%, 54%, and 38%, respectively, compared to non-inoculated plants after 3 weeks of growth. Interestingly, the results of the physiological characteristics of isolates showed that Indole-3-acetic acid (IAA) production by JW191 was the lowest among our candidate strains [13]. Thus, the strain JW191 may contribute to plant growth promotion through other mechanisms independent of phytohormone secretion. We found that reactive oxygen species ROS-associated metabolic processes were enriched, as exemplified by 6 up-regulated genes: Haem peroxidase family protein: POD2 (Os01g0326300), PRX22 (Os01g0963000), PRX45 (Os03g0368900), PRX61 (Os04g0688300), PRX65 (Os05g0134400), and PRX125 (Os10g0109300). Those antioxidant genes are involved in the response to oxidative stress and detoxification processes by scavenging ROS in plants under abiotic stress conditions. Similarly, it was reported that EPS-producing bacteria can induce the up-regulation of genes encoding antioxidant enzymes, such as superoxide dismutases and peroxidases in plants [31]. We also analyzed the relative gene expression of ascorbate oxidase gene *OsAAO5* (OS09G0507300), which was 2 times higher in rice seeds treated with EPSs compared to uninoculated seeds. Ascorbate acts as an antioxidant, regulating the redox state of the apoplast by scavenging reactive oxygen species (ROS) that accumulate under abiotic stress conditions in plants. In addition to its antioxidant function, ascorbate also acts as a cofactor for enzymes involved in the synthesis, metabolism, and modification of various substances that affect plant stress responses. Ascorbate oxidase genes influence these processes by regulating ascorbate availability. This may contribute to the stress tolerance of plants [32,33,34]. Additionally, our RNA-seq data revealed the upregulation of lipoxygenase genes LOX1 (Os03g0700700) and LOX3 (Os03g0699700). Lipoxygenase is crucial for the synthesis of jasmonic acid (JA), a compound that accumulates in plants during salt stress, serving as a key activator of stress-responsive genes [35]. Lipoxygenases also play a crucial role during seed germination and further plant growth stages [36].

Additionally, EPS treatment was found to modulate some germin-like proteins GLP, such as GLP8-7 (Os08g0189600) and GLP3-7 (Os03g0804500), with positive fold changes of 2.3 and 2.2, respectively. In general, GLPs contribute to plant development as they are involved in seed germination, root growth, and flowering processes. Furthermore, they are closely associated with plant defense against biotic stresses such as fungal pathogen invasion. GLPs also play a role in abiotic stress tolerance, such as response to salicylic acid, hydrogen peroxide, drought, and salt stress [37,38,39]. Similarly, Manosalva et al. (2009) demonstrated the important role of the GLP family in disease resistance in rice and other cereals like barley and wheat [40]. In a further study, Liu et al. (2016) reported that the GLP gene OsGLP2-1 is involved in panicle blast and bacterial blight resistance in rice [41].

Interestingly, our RNA-seq analysis revealed that EPS treatment increased the expression of several transcription factors genes such as Phytoclock 1 PCL1 (Os01g0971800), transcriptional factor B3 family protein ABIVP1 (Os03g0184500), light-related transcription factor OsGLK1 (Os06g0348800), and NAC transcription factor NAC60 (Os12g0610600). Transcription factors (TFs) are pivotal in signal transduction networks, orchestrating the translation of stress signals into the activation of stress-responsive genes. With their multifunctional nature, TFs can govern multiple pathways concurrently during plant stresses, thus serving as potent instruments for modulating regulatory and stress-responsive pathways [42]. It was found that TFs belonging to families like AP2/EREBP, MYB, WRKY, NAC, and bZIP are involved in regulating various abiotic stress responses in plants [43]. They can coordinate the activation of genes involved in osmotic adjustment, antioxidant defense, ion homeostasis, and other protective mechanisms under stress conditions [44,45].

We also found that EPS inoculation of rice seeds led to the down-regulation of heat stress-related genes like heat shock protein HSP18 (Os01g0184100) and HSP26 (Os03g0245800) and heat stress transcription factor B2C HSFB2C (Os09g0526600), which is similar to heat shock factor protein 3 HSF3. The DnaJ domain protein C53 gene *OsDjC53* (Os06g0195800), which is similar to DNAJ heat shock N-terminal domain-containing protein, was also down-regulated. In general, heat shock proteins (HSPs) act as molecular chaperones, helping to maintain the protein structure and function under biotic and abiotic stress conditions, especially heat stress. They are known to prevent protein aggregation and help maintain cellular homeostasis. Moreover, HSPs are required to protect the photosynthetic machinery and maintain the folding of proteins involved in various metabolic pathways [46,47]. In several studies, heat shock protein (HSP) genes showed differential expressions by up- or down-regulation under several abiotic stresses. For example, HSP18.1 was induced in pea plants under high-temperature stress and in grape plants under low-temperature stress. Similarly, increased expression was reported for HSP26 in wheat under high-temperature stress, in maize under drought stress, and in soybeans under heavy metal stress, like cadmium [48]. However, the expression of other HSPs was shown to be down-regulated under low-temperature stress, such as HSP90 in wheat and HSP60 and HSP21 in sunflower [49]. Thus, the modulation of heat stress-responsive genes under stress conditions in plants is a complex process that may depend on the plant species and their development stages or even on the stress length and experimental conditions.

According to our findings, we suggest that the down-regulation of genes related to heat stress in rice is part of a nuanced strategy to balance energy conservation, maintain cellular homeostasis, and allocate resources for growth, ultimately contributing to the plant’s ability to endure and recover from heat stress. We believe that by down-regulating genes associated with heat stress, the plant can conserve energy that would otherwise be expended on stress responses. As high temperatures can lead to the denaturation of proteins or disrupt their structure and function, down-regulating certain genes may help prevent the synthesis of proteins that are particularly sensitive to heat, which will reduce the risk of protein denaturation. This allows the plant to maintain the functionality of essential proteins, including those involved in growth processes. The summary of the transcriptome analysis is shown in Figure 5.

## 5. Conclusions

In summary, plant bioassays showed a general enhancement of plant growth under stress conditions by EPS inoculation. Importantly, based on the RNA-seq transcriptome analysis of rice seeds, we demonstrated that EPSs can influence the expression of a plethora of genes. Among those genes, several are known to be associated with stress responses in plants. Enhancing plant growth by EPSs may help identify strategies for developing crops that are tolerant to abiotic stress. Our data also suggest the potential to develop EPSs purified from *Enterobacter ludwigii* (JW191) as a biostimulant for rice. This study brings attention to intriguing concepts about using EPSs as a plant growth promoter under abiotic stress while prompting inquiries for future investigations.

## Figures and Tables

**Figure 1 microorganisms-12-01212-f001:**
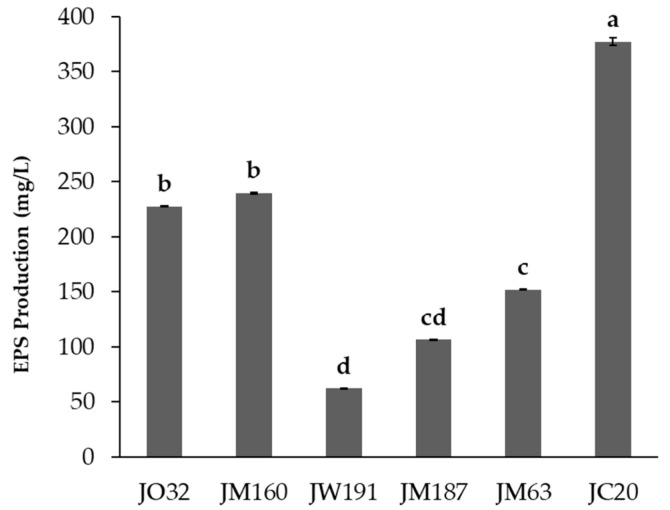
EPS production (mg/L) of selected isolates. Different letters indicate significant differences at *p* < 0.05 with Tukey’s test.

**Figure 2 microorganisms-12-01212-f002:**
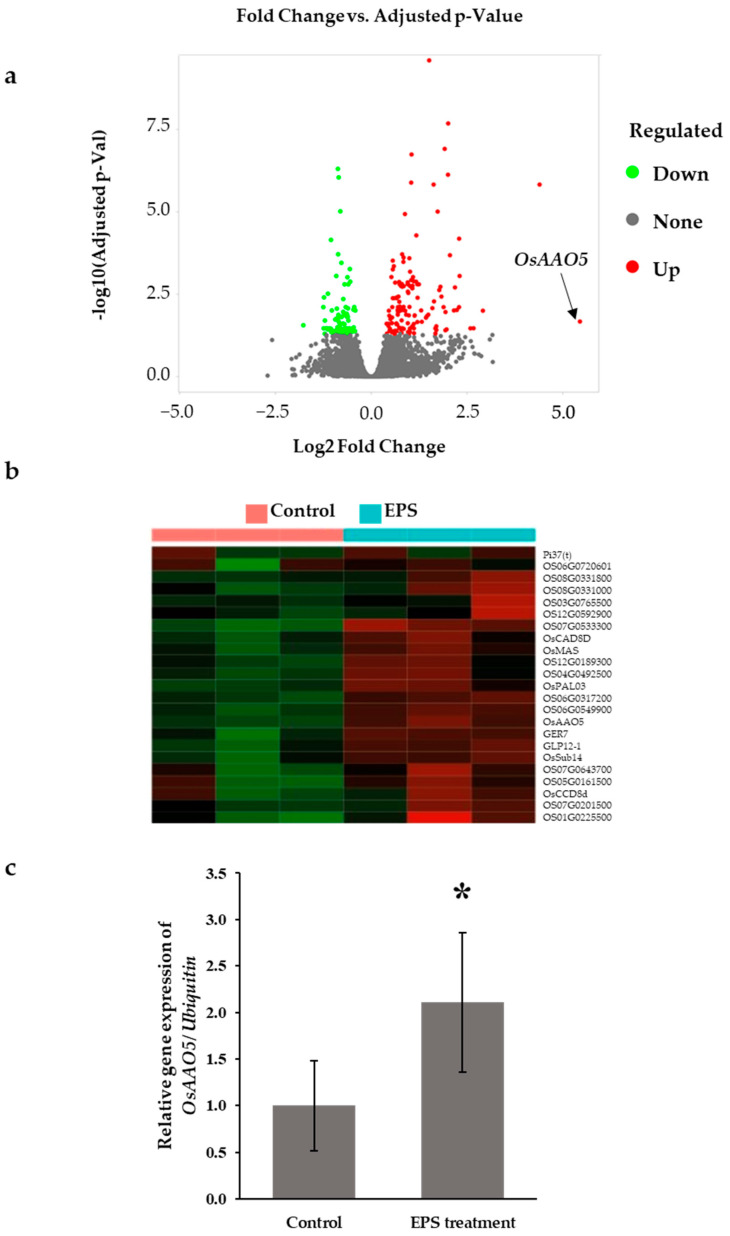
Differential expression analysis. (**a**), Volcano plot for differential expression analysis using DESeq2. (**b**), Cluster 4 of K-means clustering represents the group involved in EPS treatment. (**c**), RT-qPCR analysis of *OsAAO5* expression in rice seeds (*t*-test, * *p* < 0.05).

**Figure 3 microorganisms-12-01212-f003:**
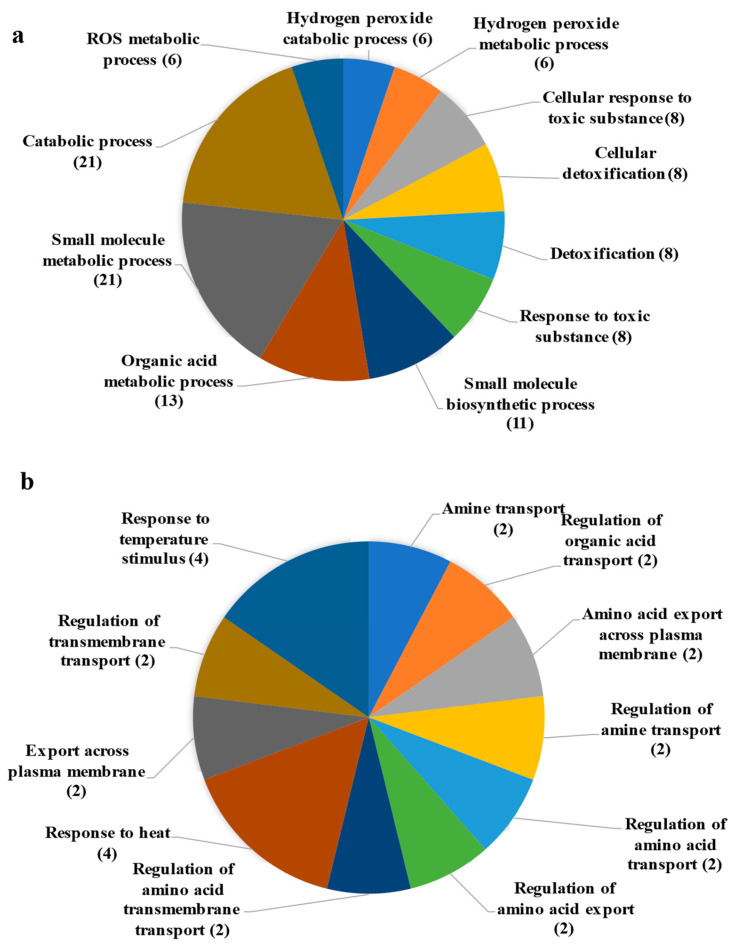
GO terms associated with biological processes in rice. (**a**) Up-regulated DEGs GO_BP analysis. (**b**) Down-regulated DEGs GO_BP analysis.

**Figure 4 microorganisms-12-01212-f004:**
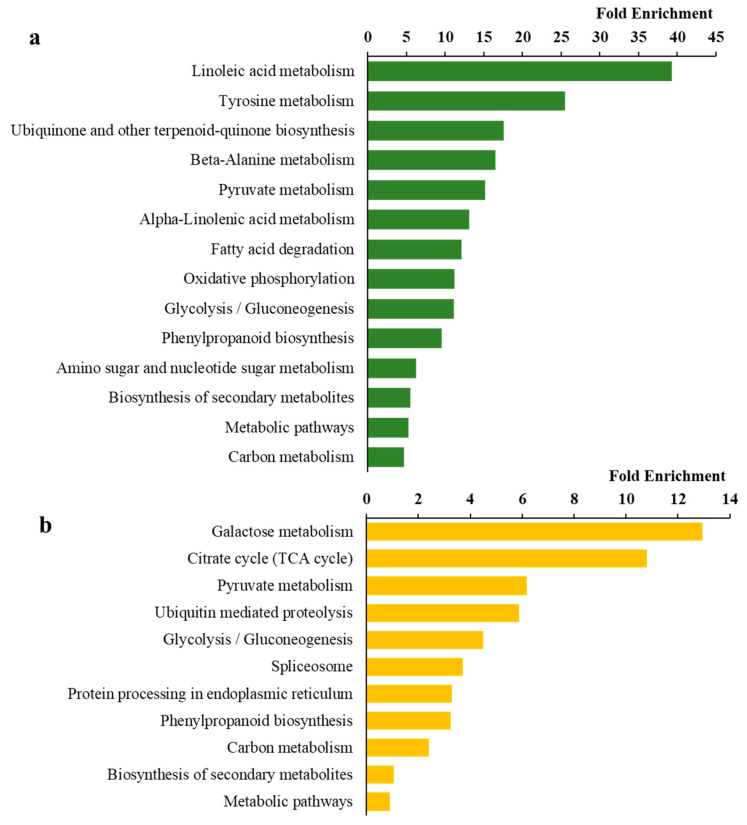
KEGG pathway analysis of differentially expressed genes. (**a**) Up-regulated genes KEGG pathway analysis. (**b**) Down-regulated genes KEGG pathway analysis.

**Figure 5 microorganisms-12-01212-f005:**
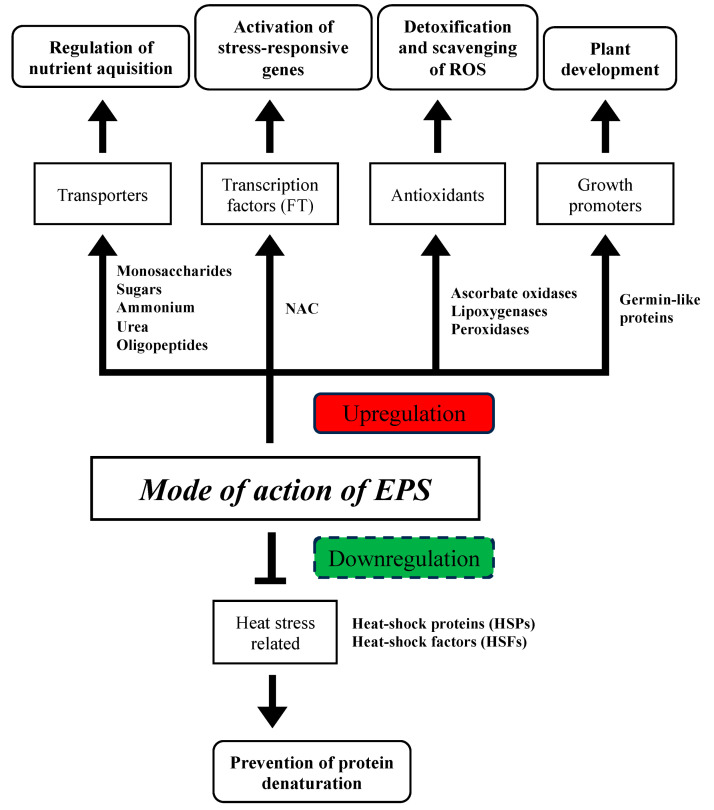
Proposed summary of the mode of action of EPSs in regulating stress tolerance in rice.

**Table 1 microorganisms-12-01212-t001:** Screening of culture media for growth and EPS production of different strains.

Isolate Name [13]	Closest Relative Based on 16S rRNA Gene Sequence [13]	Plant Rhizosphere Soil Used as Inoculant [13]	Origin of Isolates Associated with Rice Plant [13]	Production of EPS (Fluorescence Assay)				OD (600 nm)	
				TSB		YM		TSB	YM
				−G ^†^	+G	−G	+G		
JRy205	*Pseudomonas mandelii*	Rye Grass	Rice root	−	−	−	−	1.268 *	0.043
JR5	*Rhizobium daejeonense*	Rice	Rice root	−	−	+	−	0.901 *	0.083
JR207	*Stenotrophomonas rhizophila*	Rice	Rice root	−	−	−	−	0.205 *	0.026
JR172	*Agrobacterium tumefaciens*	Rice	Rice root	−	−	−	−	1.466 *	0.041
JM160	*Enterobacter* sp.	Maize	Rice leaf	−	−	−	+	1.881	1.699
JW191	*Enterobacter ludwigii*	Wheat	Rice root	−	+	−	+	1.894	1.966
JM187	*Enterobacter cancerogenus*	Maize	Rice leaf	−	−	−	+	2.108	2.022
JM75	*Pseudomonas putida*	Maize	Rice root	−	−	−	−	1.831	1.741
JR198	*Bacillus altitudinis*	Rice	Rice root	−	−	−	−	1.623 *	2.019
JC20	*Enterobacter asburiae*	Grab grass	Rice leaf	−	−	+	+	1.980 **	0.028
JO171	*Rhizobium daejeonense*	Oat	Rice leaf	−	−	−	−	1.030 *	2.061
JO32	*Enterobacter cloacae*	Oat	Rice root	−	−	−	+	1.878 **	0.036
JM51	*Agrobacterium tumefaciens*	Maize	Rice leaf	−	−	−	−	1.158 *	1.671
JM52	*Bacillus pumilus*	Maize	Rice root	−	−	−	−	1.454 **	0.005
JC100	*Xanthomonas axonopodis ASA*	Grab grass	Rice leaf	−	−	−	−	0.922 *	1.565
JR4	*Bacillus altitudinis*	Rice	Rice root	−	−	−	−	1.722 *	2.023
JM63	*Enterobacter ludwigii*	Maize	Rice root	−	−	−	+	1.957 **	0.043

^†^: 3% (*w*/*v*) glucose. The significance level was determined at the 5% level, with * and ** indicating *p* < 0.05 and *p* < 0.01, respectively, with *t*-test.

**Table 2 microorganisms-12-01212-t002:** Effect of EPS concentrations on biomass among the different strains under heat stress.

Strains	EPS Amount (µg/mL)	SL (cm/plant) before Stress	SL (cm/plant) after Stress	SFW (mg/plant)	RFW (mg/plant)	SDW (mg/plant)	RDW (mg/plant)	SPAD Value before Stress	SPAD Value after Stress
Control	0	10.4 ± 0.4 ^e^	14.1 ± 0.7 ^f^	83.0 ± 12.0 ^a^	108.1 ± 28.6 ^a^	21.6 ± 3.3 ^ac^	17.5 ± 2.5 ^ad^	27.9 ± 2.8 ^b^	20.9 ± 2.4 ^b^
JW191	1	11.7 ± 0.6 ^ae^	14.5 ± 0.1 ^df^	77.5 ± 10.0 ^a^	78.9 ± 11.8 ^a^	16.4 ± 1.2 ^bc^	10.8 ± 0.9 ^cd^	31.4 ± 1.7 ^ab^	27.8 ± 3.5 ^ab^
	10	11.8 ± 0.3 ^ad^	16.3 ± 0.6 ^af^	95.0 ± 2.9 ^a^	107.7 ± 3.5 ^a^	18.4 ± 0.6 ^ac^	14.1 ± 0.5 ^ad^	30.4 ± 2.8 ^ab^	29.4 ± 3.1 ^a^
	100	11.6 ± 0.2 ^ae^	14.1 ± 1.2 ^f^	78.5 ± 10.7 ^a^	87.6 ± 16.4 ^a^	15.5 ± 2.2 ^c^	11.0 ± 1.0 ^cd^	31.4 ± 0.4 ^ab^	27.2 ± 2.6 ^ab^
JO32	1	11.4 ± 0.5 ^ce^	14.9 ± 2.1 ^bf^	79.9 ± 15.6 ^a^	88.0 ± 17.5 ^a^	16.7 ± 3.4 ^ac^	12.6 ± 2.4 ^ad^	33.4 ± 0.8 ^ab^	27.5 ± 3.2 ^ab^
	10	11.5 ± 0.2 ^be^	15.6 ± 0.7 ^af^	84.9 ± 9.1 ^a^	76.6 ± 12.8 ^a^	16.7 ± 2.6 ^ac^	10.3 ± 2.0 ^ad^	31.8 ± 0.2 ^ab^	29.0 ± 3.5 ^a^
	100	11.7 ± 0.1 ^ae^	14.5 ± 0.7 ^df^	79.4 ± 9.8 ^a^	81.6 ± 11.3 ^a^	15.9 ± 1.6 ^c^	12.1 ± 1.1 ^bd^	31.3 ± 0.9 ^ab^	28.9 ± 1.6 ^a^
JM160	1	12.6 ± 0.2 ^ac^	16.4 ± 0.9 ^af^	90.5 ± 12.1 ^a^	95.0 ± 25.4 ^a^	18.5 ± 3.1 ^ac^	14.8 ± 2.6 ^ad^	33.0 ± 1.1 ^ab^	29.0 ± 0.8 ^a^
	10	12.5 ± 0.3 ^ac^	16.2 ± 0.1 ^af^	90.7 ± 3.1 ^a^	93.6 ± 8.7 ^a^	18.5 ± 0.2 ^ac^	13.6 ± 2.7 ^ad^	32.0 ± 1.1 ^ab^	29.6 ± 3.1 ^a^
	100	12.6 ± 0.4 ^ac^	17.8 ± 0.1 ^a^	103.7 ± 6.0 ^a^	112.4 ± 20.8 ^a^	21.2 ± 1.3 ^ac^	18.8 ± 4.5ac	31.8 ± 0.7 ^ab^	31.4 ± 1.4 ^a^
JM63	1	12.9 ± 0.1 ^ab^	16.9 ± 0.7 ^ae^	102.2 ± 3.7 ^a^	113.1 ± 6.4 ^a^	19.7 ± 1.0 ^ac^	15.8 ± 2.8 ^ad^	32.6 ± 1.4 ^ab^	30.0 ± 2.1 ^a^
	10	12.5 ± 0.3 ^ac^	16.3 ± 1.9 ^af^	98.8 ± 16.6 ^a^	112.4 ± 14.4 ^a^	20.3 ± 4.1 ^ac^	14.4 ± 2.2 ^ad^	31.7 ± 1.2 ^ab^	29.5 ± 1.2 ^a^
	100	13.0 ± 0.5 ^a^	17.6 ± 0.7 ^ab^	101.4 ± 2.7 ^a^	108.1 ± 7.8 ^a^	19.2 ± 0.8 ^ac^	13.2 ± 2.3 ^ad^	31.6 ± 2.1 ^ab^	29.2 ± 1.3 ^a^
JM187	1	10.6 ± 0.3 ^de^	14.2 ± 0.3 ^ef^	85.2 ± 3.2 ^a^	125.8 ± 35.0 ^a^	21.5 ± 0.8 ^ac^	20.1 ± 5.1 ^ab^	31.5 ± 2.1 ^ab^	28.0 ± 3.4 ^ab^
	10	10.6 ± 0.8 ^de^	14.6 ± 1.1 ^cf^	84.8 ± 11.5 ^a^	120.0 ± 17.6 ^a^	22.6 ± 2.1 ^ab^	20.5 ± 3.8 ^a^	32.2 ± 1.7 ^ab^	28.6 ± 1.9 ^a^
	100	10.4 ± 0.6 ^e^	14.7 ± 0.2 ^bf^	90.7 ± 2.9 ^a^	127.8 ± 6.4 ^a^	23.0 ± 1.0 ^a^	20.5 ± 0.7 ^a^	32.6 ± 2.9 ^ab^	27.2 ± 0.5 ^ab^
JC20	1	12.7 ± 0.6 ^ac^	17.3 ± 1.0 ^ac^	98.9 ± 7.4 ^a^	106.6 ± 18.5 ^a^	19.7 ± 2.0 ^ac^	13.4 ± 1.9 ^ad^	33.7 ± 3.9 ^ab^	29.8 ± 4.4 ^a^
	10	12.1 ± 0.9 ^ac^	16.5 ± 0.8 ^af^	94.3 ± 5.0 ^a^	110.2 ± 8.9 ^a^	19.1 ± 0.9 ^ac^	16.1 ± 0.5 ^ad^	35.5 ± 2.5 ^a^	31.5 ± 0.6 ^a^
	100	12.8 ± 0.3 ^ac^	17.2 ± 0.4 ^ad^	98.9 ± 12.6 ^a^	108.1 ± 19.9 ^a^	19.4 ± 2.5 ^ac^	15.6 ± 4.5 ^ad^	32.3 ± 1.3 ^ab^	30.8 ± 0.8 ^a^

NB: The same letters within a column indicate no significant difference at a 95% probability level at the *p* < 0.05 level with Tukey’s test, respectively. SL: shoot length; SFW: shoot fresh weight; RFW: root fresh weight; SDW: shoot dry weight; RDW: root dry weight.

**Table 3 microorganisms-12-01212-t003:** Improvement of plant growth by EPS of JW191 under abiotic stresses.

Treatment		Plant Establishment	SL (cm/Plant)	SFW (mg/Plant)	RFW (mg/Plant)	SDW (mg/Plant)	RDW (mg/Plant)	SPAD Value
Non-stress	Control	76.0 ± 18.2	22.7 ± 2.3	120.2 ± 15.5	89.2 ± 19.0	27.6 ± 4.3	15.6 ± 1.5	30.0 ± 5.1
	+EPS	100.0 ± 0.0 **	24.7 ± 2.6 *	141.3 ± 19.0 *	133.9 ± 20.7 **	30.7 ± 4.9	16.5 ± 1.0	35.8 ± 1.2 *
Drought	Control	88.0 ± 8.4	17.5 ± 0.4	39.8 ± 7.9	44.2 ± 7.4	20.1 ± 1.5	10.8 ± 2.2	5.1 ± 2.3
	+EPS	98.0 ± 4.5 *	18.4 ± 1.9	40.5 ± 5.1	58.6 ± 15.9	23.0 ± 2.6 *	13.7 ± 1.9 *	9.7 ± 1.4 **
Heat	Control	72.0 ± 17.9	16.7 ± 0.8	68.8 ± 6.4	58.3 ± 10.2	17.7 ± 2.0	9.9 ± 1.9	14.4 ± 4.1
	+EPS	100.0 ± 0.0 **	18.0 ± 0.6 **	82.8 ± 6.1 **	85.6 ± 11.1 **	21.7 ± 1.8 **	12.6 ± 1.4 *	21.6 ± 3.7 **
Salinity (150mM)	Control	88.0 ± 11.0	14.6 ± 2.2	67.2 ± 20.0	47.4 ± 9.0	18.5 ± 4.0	7.3 ± 1.5	17.6 ± 7.1
	+EPS	98.0 ± 4.5 *	14.9 ± 1.4	62.6 ± 12.6	63.2 ± 22.6	20.3 ± 3.4	7.1 ± 1.0	28.0 ± 6.2 *

* and ** indicate significant differences at *p* ≤ 0.05 and 0.01 levels in *t*-test between control and EPS inoculation in each treatment, respectively. *n* = 5. SL: shoot length; SFW: shoot fresh weight; RFW: root fresh weight; SDW: shoot dry weight; RDW: root dry weight.

## Data Availability

The original contributions presented in the study are included in the article/Appendix A, further inquiries can be directed to the corresponding authors. The RNA-seq data presented in this study is openly available in BioProject accession number PRJDB18154 at the DDBJ database.

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
