# Peer review of "Effect of Bacterial Extracellular Polymeric Substances from Enterobacter spp. on Rice Growth under Abiotic Stress and Transcriptomic Analysis"

_microorganisms, 2024, doi:10.3390/microorganisms12061212_

Round 1
Reviewer 1 Report
Comments and Suggestions for Authors
The manuscript is well-prepared, and the idea is very interesting, but I have some questions that could help improve it.
- The title needs to include the PGPR used and the crop on which the trial was conducted.
- The abstract is well-introduced, but the authors missed adding numerical data. It's only in the last sentence that I learned about the PGPR used.
- The keywords are the same as those in the title. Please change them.
Introduction:
- Is it necessary to capitalize each word in "Global Biostimulants Market"? There are many similar issues in the introduction; please resolve them.
Materials and Methods:
- Provide more information about the environmental characteristics of the site where the soil samples or plants were sampled.
- "Three plants were utilized for each treatment" is unclear. Were only three used for inoculation and as controls? Isn't that a very small number to work with? Please explain.
- For accurate species identification, comprehensive molecular characterization is essential. I strongly recommend confirming bacterial identification by including sequences of at least two protein-coding marker genes (such as TEF1α, COX1, PBR2, ACT, TUB2, REC, MDH, GAP, DNAX, or RPL10) in addition to the 16S rRNA.
- Statistical analysis: Considering the low number of experimental units used, ANOVA is not an appropriate methodology for analyzing variance. Were the assumptions checked?
Results:
- Figures and tables captions are incomplete. There are many abbreviations that need to be written out to be more readable for the readers. Additionally, to ensure clarity and accuracy, it is important to include information about the standard errors and their signs on the columns.
- Was the data for RT or fold change analyzed by ANOVA?
- Figure 3 data and caption need to be revised.
- The conclusion is very strange. Figure 5 ?????.
- Supplementary materials were included in the manuscript. They were submitted individually.
- The references are not accurate according to the journal's style.
Author Response
Thank you very much for taking the time to review this manuscript. Please find the detailed responses below and the corresponding revisions or corrections highlighted changes in the re-submitted files.

Reviewer 2 Report
Comments and Suggestions for Authors
The manuscript microorganisms-3046103 is dedicated to the comparison of the efficiency of bacterial extracellular polymeric substances isolated from selected strains on the growth and development of rice under abiotic stresses. In my opinion, the manuscript summarizes a good experimental study and presents the original results, which will be interesting to readers.
I recommend the manuscript be published in Microorganisms after minor corrections.
Comments:
- Use the correct citation style of reference in line 255.
- Check the text size in Figs. For example, Figs. 1, 2b, or 5.
Author Response

(The authors gave the same response as above.)

Reviewer 3 Report
Comments and Suggestions for Authors
The work, Elucidation of the Effect of Purified Bacterial Extracellular Polymeric Substances on Improving Plant Growth Under Abiotic Stress and Associated Transcryptomic Analysis, is very interesting and novel.
Some minor points for the authors' attention:
Line 96: Change "Ortega et al" to "Ortega et al. [15]". And remove "[15]" from the end of the sentence.
Line 99: Use the appropriate symbol for degrees Celsius "°C".
Line 162:
Change "DEGs" to "(DEGs)".
Line 210:
Change "28°" to "28 °C".
Table 2, and Table 3:
The table footer must indicate the meaning of SL, SFW, RFW, SDW, RDW.
line 255:
Correct citation "Habibi et al (2014)" to "Habibi et al. [14]".
It is recommended that the Tukey test be used in the comparison of means in Table 3.
Author Response

(The authors gave the same response as above.)

Reviewer 4 Report
Comments and Suggestions for Authors
The paper
“Elucidation of the Effect of Purified Bacterial Extracellular Polymeric Substances on Improving Plant Growth Under Abiotic Stress and Associated Transcriptomic Analysis”
gave strong evidences of the beneficial effects of EPS from six strains on rice growth under heat stress, and one strain belonging to Enterobacter ludwigii showed positive effects in other stress conditions (drought and salinity). Also, authors showed using transcriptome analysis that EPS induced several genes related to embryos response to stress.
The authors did a good work. However, I suggest for the title:
“Effect of Bacterial Extracellular Polymeric Substances on rice Growth Under Abiotic Stress and Transcriptomic Analysis of responsive genes.
In addition, some clarifications are needed in order to improve the MS quality.
Line 82: to which genera belong those strains?, please to show the seven genera.
Line 93: please to write as "…microscopy (Olympus..."
Line 95: please to show in the subheading that experiment involve also measurement of EPS.
Line 96: please to give number of strains subjected to EPS extraction.
Line 107: please to give how EPS measurement was expressed (unit).
Line 114 correct as "4°C".
Line 121: please to give soil physic-chemical characteristics.
Line 123: please to give number of bacteria; origin of the EPS.
Line 131: please to give how plant establishment was evaluated, and reference.
Line 132: please to show that this is a measurement for chlorophyll (SPAD values).
Line 134: did authors extract RNA from seeds subjected also to heat or another kind of stress?
Line 138: please to correct “sterilized” by "disinfected" or "surface sterilized."
Line 139: were the seeds incubated in Petri dishes, please to give experiment with details?
Line 140: please to give seeds number for each bacterial EPS treatment and control.
Line 184: It not clear whether this experiment was performed in the presence of bacterial EPS, please to show that experiment was conducted with only one species, also how the plants were treated by EPS, and at which concentration.
2.7 Please to amend the subheading by indicating that plant assays were performed with EPS from the species “Enterobacter ludwigii” or “strain JW191”.
Line 186: It is not clear whether those seeds were treated by bacterial EPS?, please to check for clarity.
Line 221: As the authors identified the strains in table 1, please to give name of the selected most effective species and label between parenthesis, at least once in the text. In addition, in the conclusion section, please to write (JW191) just after the species name.
Line 131: please to present the figure1 after the text.
Line 23:: for the table and whole MS: please to homogenize writing for "mL".
Line 255: please to show the species name.
Line 255-261: This part are results is obviously from the previous subheading. In addition some elements are of “Discussion”. and should be placed in it. Please check or explain.
Line 256: please to present briefly those changes.
Line 338: delete "t".
Line 339: please to give results as values, especially for highest negative effects of stresses and positive changes with EPS.
Line 346: overall, give values for comparisons.
Line 465: please to check for writing.
Figures
Figure S2: please to give in the caption, the significance of each abbreviation presented in the body o the figure.
Figure S4 and line 283: Please to show clearly the significance of EPS1 EPS2 and EPS" in M&M, in the figure S4, please to amend the caption with significance of all abbreviation shown in the figure body.
Figure S3: and Line 277: same remark as for FigS2. Present most relevant results of FigS3a in “Result section”, In addition, please to show panel letters (a, b, c, d) on the body of each figure and write significance of abbreviations in the caption. In the text, please to present results of FigS3d.
Table 1: Results of OD do not show statistics
Author Response

(The authors gave the same response as above.)

Round 2
Reviewer 1 Report
Comments and Suggestions for Authors
After reviewing the revised version, I believe the authors have addressed my concerns well, so the manuscript can be accepted now.